# Mapping CRISPR spaceromes reveals vast host-specific viromes of prokaryotes

Sergey A. Shmakov[1], Yuri I. Wolf [1], Ekaterina Savitskaya[2,4], Konstantin V. Severinov[2,3] & Eugene V. Koonin [1✉]

CRISPR arrays contain spacers, some of which are homologous to genome segments of viruses and other parasitic genetic elements and are employed as portion of guide RNAs to recognize and specifically inactivate the target genomes. However, the fraction of the spacers in sequenced CRISPR arrays that reliably match protospacer sequences in genomic databases is small, leaving the question of the origin(s) open for the great majority of the spacers. Here, we extend the spacer analysis by examining the distribution of partial matches (matching k-mers) between spacers and genomes of viruses infecting the given host as well as the host genomes themselves. The results indicate that most of the spacers originate from the host-specific viromes, whereas self-targeting is strongly selected against. However, we present evidence that the vast majority of the viruses comprising the viromes currently remain unknown although they are likely to be related to identified viruses.

[1] National Center for Biotechnology Information, National Library of Medicine, Bethesda, MD 20894, USA. [2] Institute of Molecular Genetics, Russian Academy of Sciences, Moscow, Russia. [3] Waksman Institute of Microbiology, Rutgers, State University of New Jersey, Piscataway, New Jersey 08854, USA. [4]Deceased: Ekaterina Savitskaya. ✉email: koonin@ncbi.nlm.nih.gov

CRISPR-Cas are adaptive immunity systems of bacteria and archaea. The distinctive property of CRISPR-Cas is the ability to incorporate short segments of foreign DNA (spacers) into CRISPR arrays. The complementarity of the CRISPR (cr) RNA, transcribed from the CRISPR array, to the target site (protospacer) in the cognate foreign DNA or RNA guides target recognition and cleavage by the CRISPR-Cas interference machinery, rendering the host resistant to subsequent infections by the same or closely related agents[1–6]. A recent comprehensive survey of CRISPR spacers has shown that most of the identifiable protospacers originate from viruses, proviruses, or other mobile genetic elements[7]. Because protospacer identification relies on comparison of short (20–40 nt) nucleotide sequences, to avoid spurious matches, the search must be highly restrictive, allowing one or two mismatches at most. Under this strict criterion, search of the available genomic databases resulted in the detection of protospacers for less than 10% of the CRISPR spacers[7]. At least three, not necessarily mutually exclusive hypotheses on the origin of the "dark matter" that comprises the bulk of the spacerome, can be considered. First, the dark matter spacers could represent partial matches to viral genomes that would result, primarily, from viral mutational escape and are missed in database searches due to the strictness of the search criteria. There could be selective pressure for CRISPR arrays to keep partially matching spacers. Some of the CRISPR-Cas spacers, particularly, those from type III, are competent for interference despite multiple mismatches[8–10]. Furthermore, partially matching spacers can mediate primed adaptation, i.e. highly efficient acquisition of spacers from genomes of invading mobile genetic elements genomes following the recognition of a partially matching crRNA[11–14]. Second, the spacers without matches could originate from still unknown components of the virome (or, more generally, the mobilome). Third, spacers could come from sources other than the mobilome, in particular, from the host genome itself, having mutated to partial matches. Autoimmunity in CRISPR-Cas systems has been reported but accounts only for a small fraction of spacers with matches[7,15]. By contrast, active incorporation of self-matching spacers has been observed in experimental systems with inactivated interference machinery[16]. Thus, spacers with partial matches to the host genome could accumulate as a result of selection against autoimmunity.

Here, in an attempt to elucidate the origin(s) of the spacerome dark matter, we limited the analysis to a spacer collection from hosts matched to viruses through spacer–protospacer pairings. The distributions of partial matches for the spacers were compared to the respective distributions for mock spacers generated from the host genomes. The results reveal strong avoidance of self-matches and pronounced enrichment of the spacerome with virus matches, but inclusion of partial matches increased that enrichment only slightly. Thus, most of the dark matter spacers appear to originate from the dominant but still unknown, most likely, host-specific segment of the mobilome.

## Results

### Virus-matching and host-matching k-mers in CRISPR spacers.
Of the 2102 complete bacterial and archaeal genomes containing CRISPR arrays[6], 154 were found to carry at least one spacer with an identifiable protospacer (with at most one mismatch) in the viral sequence database (Fig. 1a). A comparison of the lengths (i.e. number of spacers) of the arrays containing spacers with identifiable virus matches with those lacking such matches yielded distributions with similar modes but with a much sharper peak for the matchless arrays (Fig. 1b). This difference might reflect greater recent activity in the match-containing arrays (see "Discussion").

Matches between spacers in CRISPR arrays and protospacer in virus genomes comprised the host–array–virus links that were used to construct the dataset for the detailed spacerome analysis. We chose to analyze these host-linked viromes in order to minimize spurious matches produced by short k-mers. Altogether, the 154 genomes linked to viruses contained 392 CRISPR arrays with 10,555 individual spacers. To match this set of spacers, an equal number of fragments with the same length distribution was randomly generated by sampling the sequences of the host genomes outside of the CRISPR arrays. For each $k$ from 8 to 22, the set of the CRISPR spacers or the randomly sampled mock spacers were transformed into non-redundant sets of $k$-mers using a sliding window with the step of 1. Then, occurrences of these $k$-mers were identified in the target viral genomes and in the corresponding host genome, excluding the source arrays (Fig. 1c). These comparisons produce four sets of hits (i.e. coordinates of the matching $k$-mer locations) for: (1) spacers in linked viruses; (2) spacers in host (self) genomes; (3) mock spacers in viruses; and (4) mock spacers in host genomes. The mock spacer set provides the benchmark for the spurious matches between the spacerome and the virome because the overwhelming majority of the host genome sequence is unrelated to the virus genomes. The mock spacers sampled from the host genome were used because, as shown previously, oligonucleotide statistics (GC-content, dinucleotide and tetranucleotide distributions) of a spacerome is similar to that of the corresponding host genome and the linked virome[7], so the random expectation is difficult to derive analytically. In all comparisons, the fraction of spacers with at least one match of the given length $k$ was used to quantify the similarity.

The collected data on host–virus affinities were used to compare the number of spacers containing at least one $k$-mer matching potential targets for the real and mock spacers, i.e. comparing the datasets 1 vs 3 and 2 vs 4. For small $k$, the lists of matches are dominated by random occurrence of identical $k$-mers, but matches of longer $k$-mers are likely to originate from homologous sequence fragments, and thus, the numbers of such matches are expected to differ substantially between the real and mock spacers. The ratio of the number of spacers with $k$-mer matches in the target virus sequences to the corresponding value obtained for the mock spacer set indicates the relative enrichment or depletion of matches (Supplementary Fig. 1). As expected, a strong enrichment of spacers matches in viruses compared to mock spacers was observed for $k \geq 10$, reaching a factor greater than 50 (Fig. 2a). The same comparison shows depletion of $k \geq 11$-mers matching host genomes by a factor of about 4 for longer $k$-mers (Fig. 2a). Self-targeting analysis for the entire set of 2102 completely sequenced genomes with CRISPR arrays (i.e., including those that had no spacer matches with viruses) shows the same trend for self-matching depletion (Fig. 2b). It should be noted that the match rate between the mock spacers and host genomes (self-targeting) did not saturate at (or near) zero, but rather stayed relatively high, with 4–5% of matching $k$-mers even for $k \geq 18$ (Fig. 2c). This occurs because genomes typically contain multiple copies of integrated mobile genetic elements and other repetitive sequences that, once sampled into the mock spacer set, produce multiple hits (Supplementary Data 1). In contrast, for viral genomes, which are effectively devoid of repeats, the matching rate for mock spacers saturated at 0.1% (Fig. 2c).

Inspection of genomes with unusually high self-targeting ratios shows that the majority of protospacers in the host genomes are located inside proviruses, with only a small fraction of spacers targeting non-proviral regions (Supplementary Data 2). Apparently, these organisms protect themselves from the self-destructive action of CRISPR-Cas system via provirus-encoded anti-CRISPR proteins, as demonstrated by previous, extensive

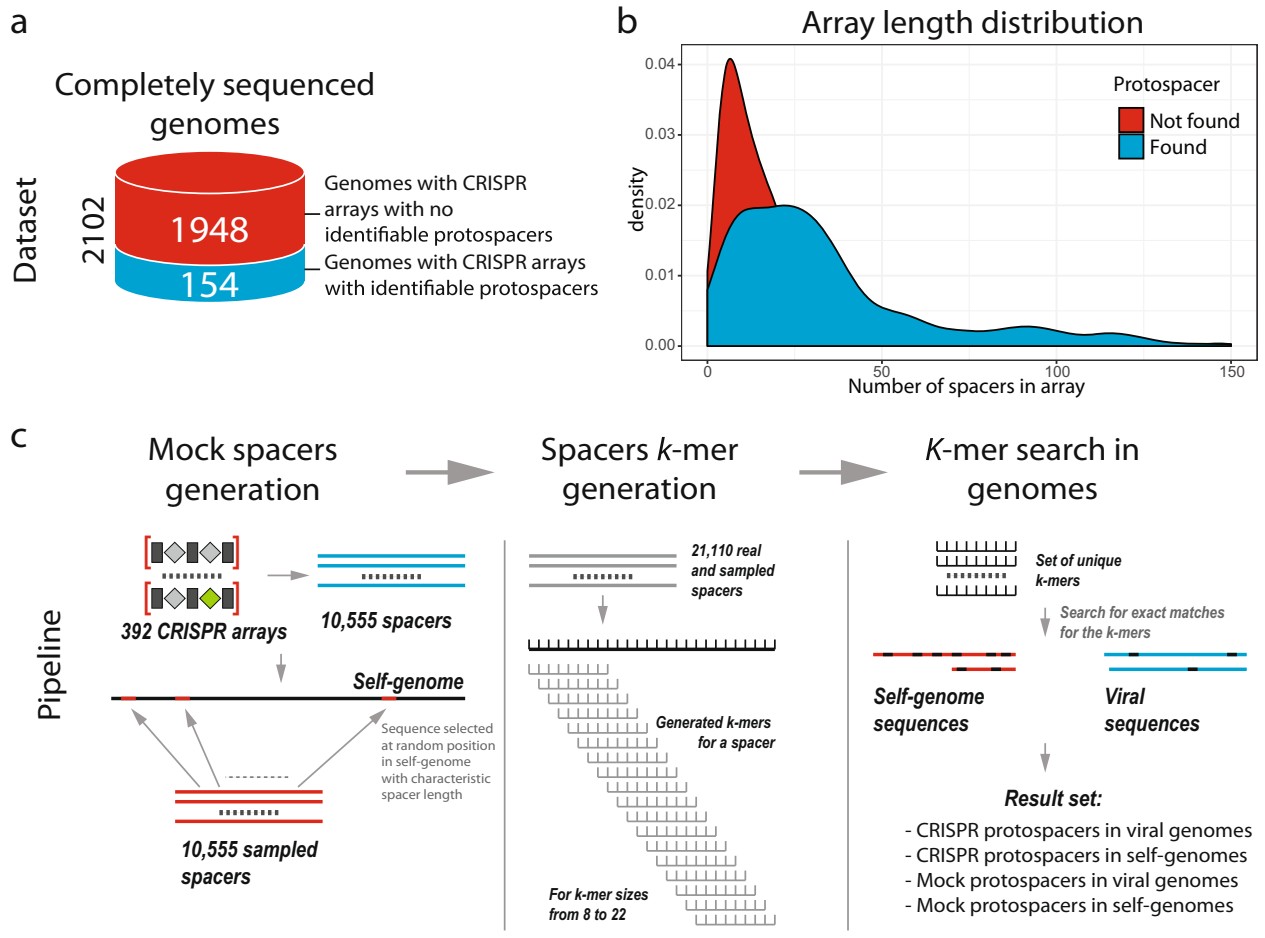

**Fig. 1 Datasets and the computational pipeline for spacer analysis. a** The analyzed genomes. On the left, the green part of the cylinder shows complete genomes containing CRISPR arrays with no identifiable protospacers, and the blue part shows complete genomes containing spacers with identifiable protospacers. The schematic on the right shows a genome with CRISPR arrays, one of which contains a virus-targeting spacer. **b** Distributions of the number of spacers per array for arrays containing protospacers with detectable matches (blue) and arrays lacking such protospacers (red). **c** The computational pipeline for spacer analysis. The workflow includes the generation of mock spacers, *k*-mer generation, and search for *k*-mer matches in spacers. The underlying data are available at ftp://ftp.ncbi.nih.gov/pub/wolf/_suppl/spacers2020/.

analyses of self-targeting in conjunction with the identification of anti-CRISPR proteins[17–20], or else, have inactivated CRISPR-Cas interference genes. Proviruses are also responsible for the non-zero rate of matches between the mock spacers and viruses (Supplementary Data 3).

It has been shown previously that the fraction of spacers with stringent (at most 1–2 mismatches per spacer) matches to viral sequences across the bacterial and archaeal CRISPR arrays was close to 7%, on average[7]. This leaves the overwhelming majority of spacers with no identified protospacer candidates. Our current observations agree with these results: when a spacer-to-virus match is defined as at least one *k*-mer from the spacer matching a viral genome, the per-spacer match rate declines from nearly 100% for $k = 8$ to 7.7% for $k = 22$. Obviously, short *k*-mers are ubiquitous and the match rate is dominated by spurious matches, whereas longer *k*-mers are unlikely to emerge independently and represent spacer acquisitions, possibly, eroded by mutations. For intermediate values of *k*, some matches are spurious, whereas others are derived from true spacer–protospacer pairs.

To assess the spacer–virus match rate under relaxed criteria (i.e., for *k*-mers that are as short as possible, but still eliminate most of the spurious matches), we took advantage of two observations to obtain the lower and upper bound estimates.

First, scrambled spacers (the sequence of each individual spacer was randomly shuffled) were generated for the set of 154 genomes and the *k*-mers generated from these scrambled spacers were matched to the virus sequences (Supplementary Fig. 2). This analysis shows that, for $k \geq 15$, the ratio between per-spacer match rates for real and scrambled spacers exceeds 100, suggesting that more than 99% of the matches are non-random. Because shuffling preserves only the nucleotide composition but destroys all higher-order autocorrelations in the sequence, this procedure is expected to underestimate the spurious match rate and, therefore, provides the upper bound for the discovery of true matches. Second, at the other extreme, the ratio between real spacer and mock spacer matches saturates at ~54 for $k \geq 18$. Mock spacers overestimate the spurious match rate due to the matches between proviruses and viruses and, therefore, provide the lower bound for the rate of true matches. Furthermore, for $k \geq 22$, the match rate remains saturated. The spacer–virus match rate was 12.0% for $k = 15$, 9.4% for $k = 18$, and 7.7% for $k = 22$. Thus, relaxing the matching criteria ratcheted up the spacer–virus match rate by a factor of 1.21–1.56, a non-negligible but modest increase. The great majority of the spacers (>88%) remain without identifiable protospacers. Another salient observation is that, at $k = 16$, the spacer match rate for host genomes becomes

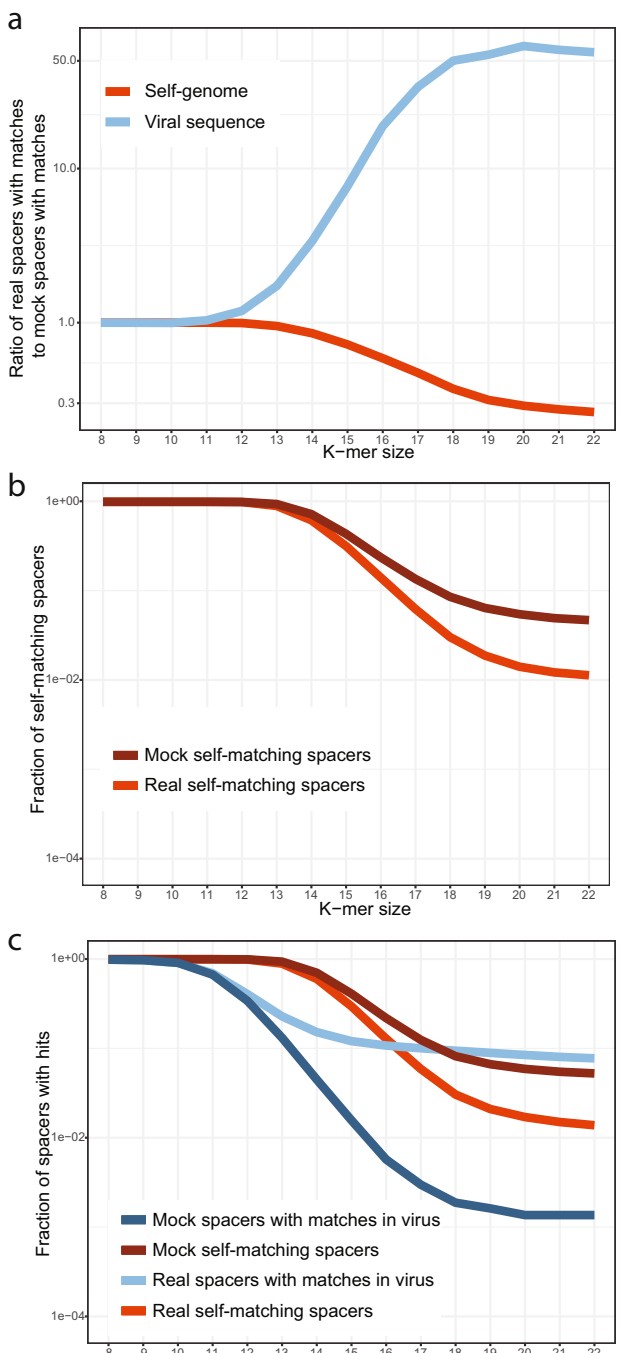

**Fig. 2 *k*-mer matches in real and mock spacers. a** Ratio of the number of *k*-mer matches for real spacers to the number of matches for mock spacers: blue, matches in viral genomes, red, matches in host genomes. The results were obtained for the dataset of 154 complete prokaryotic genomes with identifiable protospacers in viral genomes. **b** Fraction of self-matching spacers (of the total number of spacers) for real (red) and mock (dark red) spacers. The results were obtained for all 2102 complete genomes with CRISPR arrays. **c** Fractions of virus-matching and self-matching spacers (of the total number of spacers). Blue, real spacers matching viral genomes; dark blue, mock spacers matching viral genomes; red, real spacers matching the host genome; dark red, mock spacers matching the host genome. The results were obtained for the dataset of 154 complete prokaryotic genomes with identifiable protospacers in viral genomes. The underlying data are available at ftp://ftp.ncbi.nih.gov/pub/wolf/_suppl/ spacers2020/.

lower than the spacer-virus match rate, supporting the avoidance of longer self-matches. Taken together, these findings reject the hypotheses that the bulk of the dark matter of the spacerome consists of either diverged virus-derived or self-derived spacers, and thus uphold the only remaining possibility, namely, that the unmatched spacers come from the uncharted part of the mobilome.

**Virus-matching spacers in bacteria with well-studied viromes.** To further investigate the connections between the CRISPR spaceromes of specific microbes and the corresponding viromes, we selected several organisms with the largest affiliated viromes. The following sets of hosts (a subset of the 2102 genomes containing CRISPR-*cas* loci) and viruses were analyzed: 135 host and 872 virus genomes for *Escherichia*, 39 and 458 for *Pseudomonas*, 65 and 898 for *Mycobacterium*, 106 and 116 for *Streptococcus*, 26 and 263 for *Bacillus*, and 19 and 58 for *Sulfolobus*. The *k*-mer analysis was performed for each group of organisms and the corresponding virus sequence set (Fig. 3, Supplementary Fig. 3). The ratios of *k*-mer matches for the real and mock spacers showed the same avoidance of self-targeting as described above for the cumulative analysis but the enrichment of virus matches was notably less pronounced while remaining significant (Fig. 3, *p* value < $10^{-6}$; see "Methods" for details). Examination of the individual host–virome datasets showed a bimodal distribution of the virus–spacer matches. *Pseudomonas, Streptococcus*, and *Sulfolobus* showed substantial enrichment of virus *k*-mer matches along with the pronounced depletion of the self *k*-mer matches. By contrast, in *Escherichia, Mycobacterium*, and *Bacillus*, both viral and the host *k*-mer matches were depleted compared with matches obtained with mock spacers. For the latter group of bacteria, the depletion of viral matches can be explained by the absence of the actual protospacer sequences from the currently available viromes and by the high abundance of proviruses, which leads to the increase in the number of mock spacer matches (Fig. 3).

**Virus- and self-matching spacers from adaptation experiments.** The analysis described above demonstrates the avoidance of self-targeting and the enrichment of the spacerome with viral matches accumulated over unknown but, generally, long time scales. To study the short-term dynamics of spacer acquisition, we interrogated the data from laboratory spacer adaptation[21]. In these experiments, cultures of *Escherichia coli* carrying an active Type I-E CRISPR-Cas system were infected with bacteriophages T5 and λ. The CRISPR array was engineered to contain a single spacer against the respective target, with a single mismatch. As a result, 229,164 (7591 unique) and 229,877 (20,677 unique) spacers were acquired from T5 and λ, respectively. This experimental system provides the opportunity to analyze a large set of spacers for an individual virus and a specific CRISPR-Cas system, complementing the across-the-board analysis of large but sparse spacer sets described above. We performed the *k*-mer analysis for these spacer sets and compared the experimental data to the data available for Enterobacteriales (the corresponding subset of the 154 genome dataset discussed above) (Fig. 4, Supplementary Fig. 4). For both bacteriophages, a strong preference for phage-specific spacers over host-derived spacers was detected. However, in both series of phage infection experiments, the level of self-targeting by CRISPR-Cas systems was notably higher than that observed in the bulk analysis. Thanks to the known orientation of the array and the exact boundaries of the spacers in the experimental data, it was possible to analyze protospacer-adjacent motifs (PAMs) that are required for both adaptation and interference[22]. The authentic AAG PAM was observed for 99% and

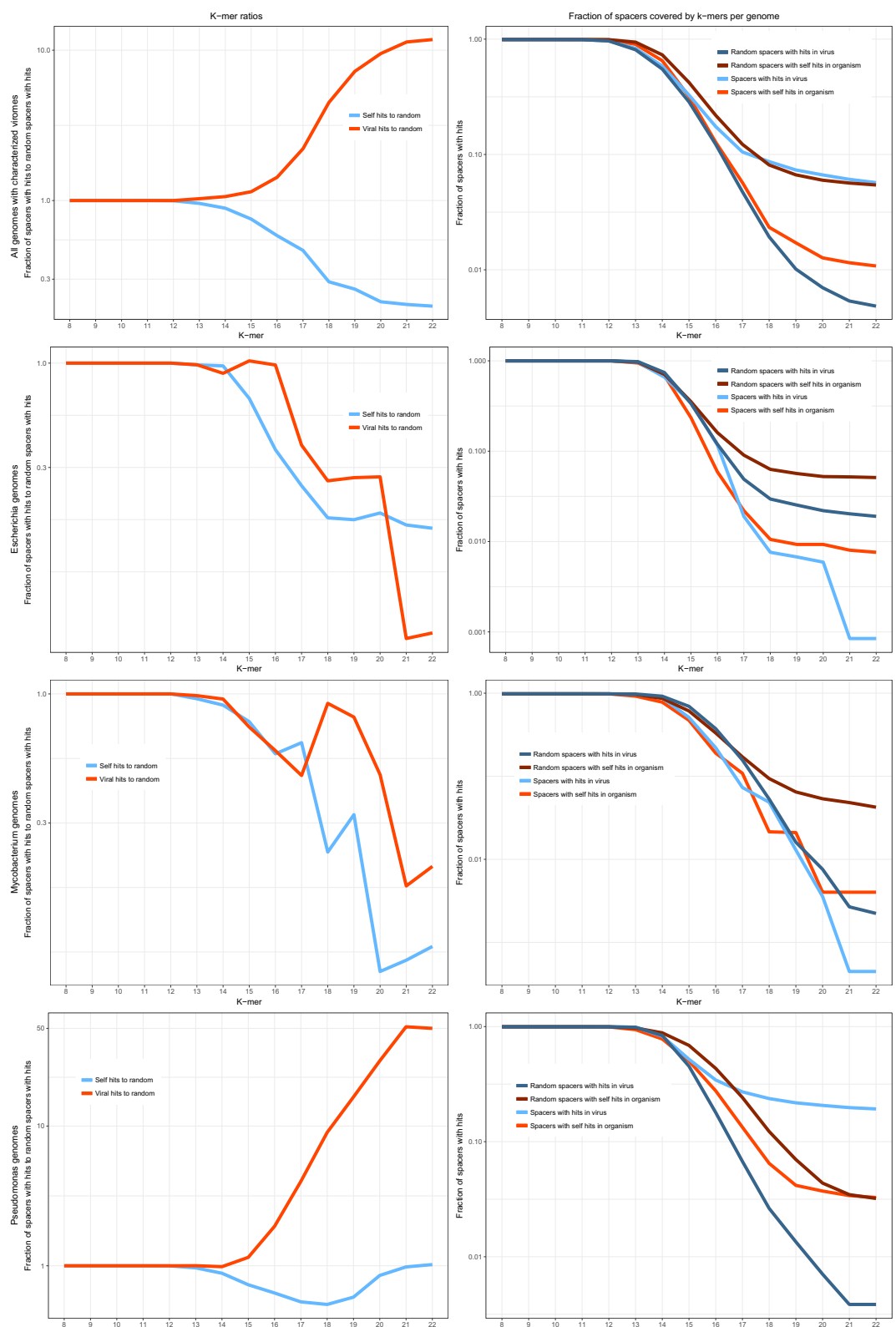

**Fig. 3** (continued)

92% of protospacers originating from the T5 and λ genomes, respectively. By contrast, AAG was found adjacent to only 60% and 73% of the self-targeting spacers acquired in these experiments (Supplementary Data 4). Such apparent sloppiness of the

self-targeting adaptation might be caused by the greater diversity of host sequences compared to the phage sequences. The difference between the self-targeting levels observed in these experiments (notwithstanding the caveat that some of the arrays in the

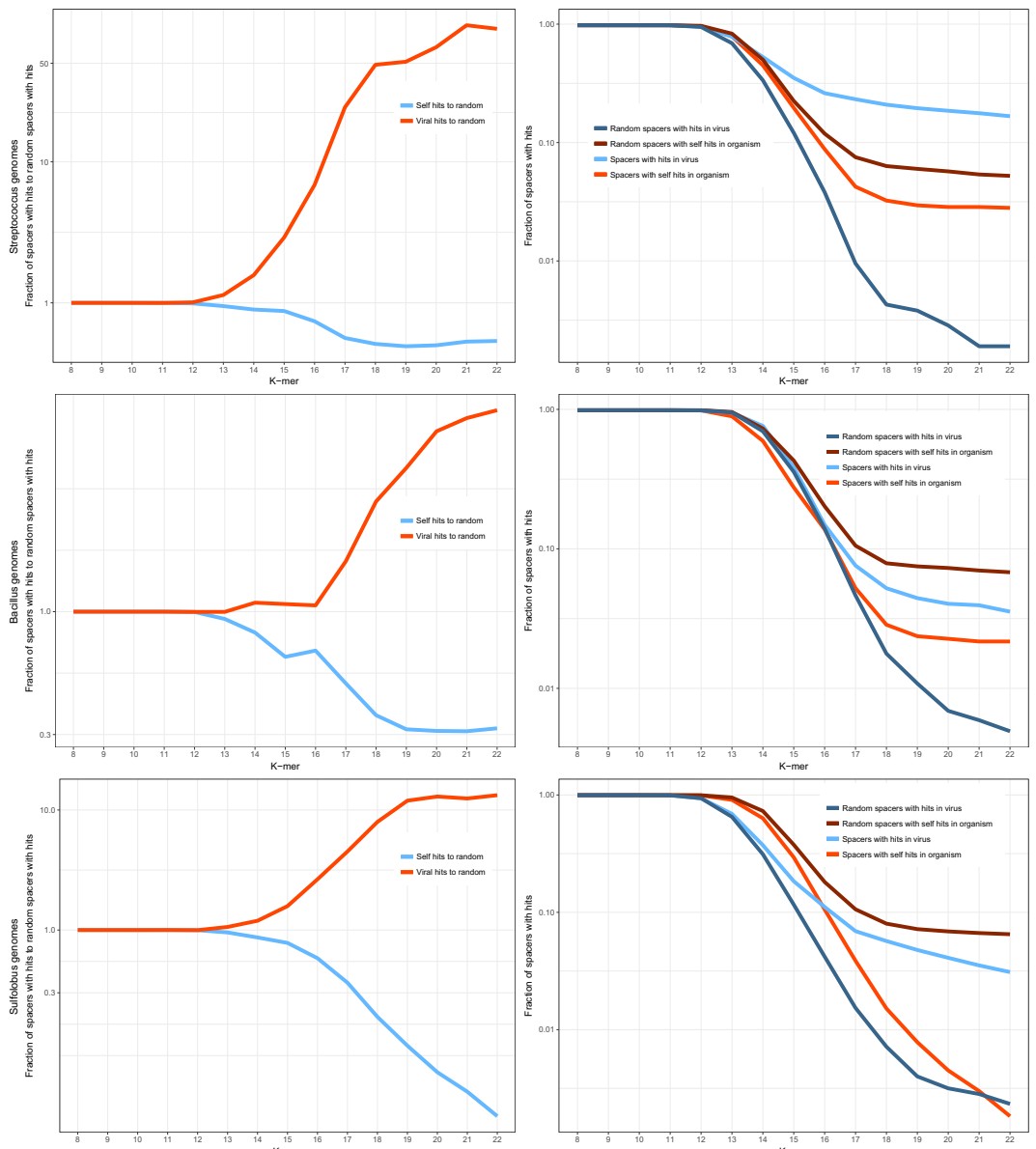

**Fig. 3 Comparison of the fractions of *k*-mer matches for real and mock spacers from prokaryotes with well-characterized viromes.** The top left panel shows a cumulative plot of the fractions of spacers with *k*-mer matches for 390 host genomes and 2665 viral genomes. Each of the other panels shows the same fractions for the individual bacteria or archaea and the associated viromes. Blue, real spacers matching viral genomes; dark blue, mock spacers matching viral genomes; red, real spacers matching the host genome; dark red, mock spacers matching the host genome. The underlying data are available at ftp://ftp.ncbi.nih.gov/pub/wolf/_suppl/spacers2020/.

experimental data might originate from dead bacteria) and those found in organisms where adaptation events accumulated over long time intervals[23] suggests that self-targeting spacers are efficiently removed by purifying selection during microbial evolution.

## Discussion

The analysis of *k*-mer matches between CRISPR spacers and virus genome compared to matches to the host genomes yields insight into the likely origin of most if not all spacers. The strong avoidance of self-targeting indicates that the majority of spacers cannot come from the host (self) genome. Conversely, the paucity of detected virus-specific spacers with mismatches shows that most of the spacers are not decaying segments of genomes of known viruses. Thus, the only remaining possibility seems to be

that the dark matter of the spacerome comes from the vast portions of the viromes (>80%) that currently remain unsampled. An accurate estimation of the size of the dark fraction of the virome requires statistical models of the virome structure and the CRISPR adaptation process which is far beyond the scope of the present work. A naïve linear approximation for the observed fraction of spacers with virus matches (about 8%) would suggest that the viromes, on average, are at least an order of magnitude greater than the portion currently sampled by the CRISPR-Cas systems. However, this is, at best, a crude lower bound for the size of the virome. There is little doubt that, statistically, the sampled viruses represent some of the most abundant members of the virome whereas the "tail" of lower abundance viruses is likely to be vast.

As shown previously, the oligonucleotide composition of the spacers closely follows those of the host genome and of known viruses infecting the respective organism[7]. Combined with the

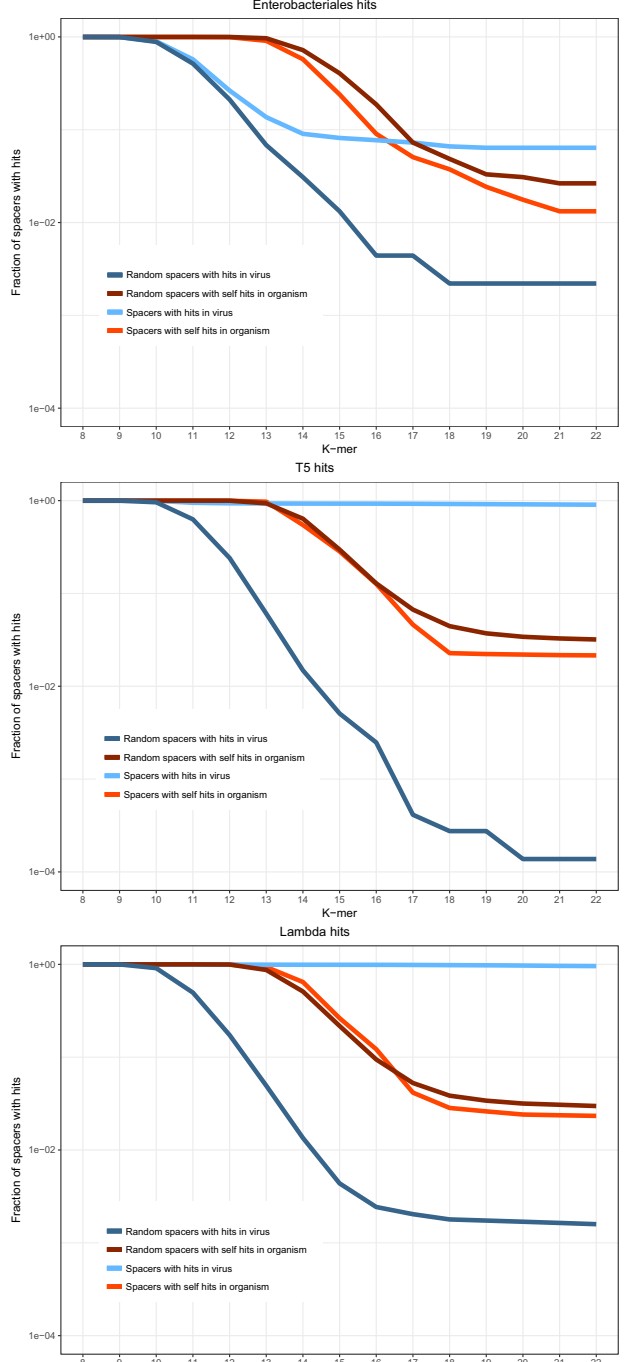

**Fig. 4 k-mer spacers matches from adaptation experiments.** The fraction of spacers with k-mer matches is shown for three datasets: 11 genomes of Enterobacteriales from the dataset of 154 genomes containing spacers with matches; *E. coli* infected with T5 bacteriophage; *E. coli* infected with λ bacteriophage. Blue, real spacers matching viral genomes; dark blue, mock spacers matching viral genomes; red, real spacers matching the host genome; dark red, mock spacers matching the host genome. The underlying data are available at ftp://ftp.ncbi.nih.gov/pub/wolf/_suppl/spacers2020/.

present observation on the paucity of spacers with mismatches, this trend indicates that the dark matter of viromes consists, primarily, of viruses that are related to the already known ones, but are diverged enough to prevent substantial cross-matching of spacers. This conclusion is compatible with the results of several

recent studies that specifically matched CRISPR arrays with the viromes from the respective habitats. In these studies, viral matches comprised a substantial fraction of spacers, in some cases, more than 50%, which is indicative of extensive sampling of viromes by the CRISPR adaptation machinery[24–26]. Furthermore, CRISPR-Cas systems appear to be major drivers of phage genome diversification[27] which can be expected to yield the type of virome with a high level of microdivergence that is inferred from our present results. It thus follows that most microbes apparently host expansive viromes that are highly habitat specific. In particular, it has been shown that human gut viromes are stable over an individual's life span but highly individual specific[28,29].

Additional analyses reported here reveal substantial spacerome variation depending on both the hosts and the viruses. The k-mer analysis of spacers from hosts with apparently well-characterized viromes (or, more precisely, numerous viruses confidently assigned to these hosts) reveals a striking dichotomy, a strong preference for virus-derived spacers in one group of prokaryotes contrasted by an apparent depletion of virus matches in another group. The depletion is likely to be an artifact caused by the prevalence of prophage sequences among the mock spacers used as a control. However, the lack of evidence of spacer acquisition from the genomes of known viruses indicates that the latter particularly poorly represent the apparently vast viromes of the respective hosts. Strikingly, the latter group includes *E. coli*, the bacterium with the largest known number of phages. In this particular case, the very small number of spacer matches (1 viral match and 17 self-matches) likely reflects the fact the subtype I-E CRISPR-Cas systems carried by *E. coli* appear to have been silent over thousands of years[23].

Analysis of experimental data on spacers acquisition demonstrates a substantially higher rate of self-targeting than estimated from the analyses of genomic databases, suggesting that purifying selection purges self-targeting spacers over many microbial generations. These findings are compatible with extremely high levels of self-targeting that has been observed in adaptation experiments with inactivated CRISPR effector nucleases[30].

Taken together, the results of this work show that the great majority of the CRISPR spacers come from the vast unsequenced part of prokaryotic viromes (mobilomes) and that these viromes are organism specific. The most abundant viruses from the viromes of well-characterized prokaryotes are likely to be already known, but the tail of less common viruses as well as virus genome micro-diversity appear to be enormous. This conclusion on the virome organization might not in itself be surprising, but to our knowledge, so far, there has been no specific evidence to support it.

## Methods

**Datasets.** Information on 2102 prokaryotic genomes containing CRISPR arrays (363,468 unique spacers) as well as the data on protospacers were obtained from a previously described dataset which was assembled in March 2019 from complete genome sequences of bacteria and archaea available in the NCBI databases[7]. Genome completeness was based on NCBI annotations[31]. Of the 2102 genomes, 154 contained spacers (10,555 unique spacers) of which at least one matched a viral sequence (no more than one mismatch). Viral genome sequences were extracted from the NT database at the NCBI[31] (sequences, classified as "Viruses" at the top of the taxonomy hierarchy) using the Entrez utilities on GenBank website.

For the analysis of genomes with well-characterized viromes, virus–host associations were identified for viruses in the NCBI databases using TAXID search. Host genomes (including all the plasmids and other extrachromosomal elements if part of the assembly) were taken from the dataset of 2102 genomes containing CRISPR-*cas* loci and grouped by genus. The six groups with the greatest number of associated viruses in the viral part of the NT database, namely, *Pseudomonas, Streptococcus, Escherichia, Mycobacterium, Bacillus, Sulfolobus* were used for further for the analysis. The selected organisms from these groups encompassed 15,277 unique spacers.

The data from adaptation experiments[21] included 229,164 spacers (7591 unique sequences) for bacteriophage T5 infection and 229,877 spacers (20,677 unique

sequences) for bacteriophage λ infection. The *E. coli* KD263 was analyzed as the host genome for this data analysis, and complete genomes of *Enterobacteria* phage T5 (NC_005859.1) and phage λ (NC_001416.1) were analyzed as viral genomes. The PAM sequences, i.e. three upstream nucleotides including the first position of the protospacer, were retrieved for the first perfect match of each spacer in these genomes and in the host genomes.

**Mock spacer sampling**. Mock spacers sets were sampled from host (self) genome sequences using information on the real spacers from the CRISPR arrays contained in the given genome. For each spacer from the CRISPR arrays, a fragment of the host genome sequence was chosen by extracting a segment of the same size as the spacer from a random genomic location. The actual CRISPR spacers were excluded from the random location selection. This procedure was applied to 10,555 unique spacers from the set of 154 genomes described above to generate 10,555 mock spacers. The same procedure was applied to the 363,468 unique spacers obtained from the entire set of 2102 complete genomes, and the other datasets described above.

**K-mer generation and search for protospacers**. Each spacer from the real and mock spacer sets described above was used to generate $k$-mers. All possible continuous subsequences of the length $k$ ($8 < k < 22$) (the probability to find an 8-mer in a host or viral genome is close to 1; 22 is the minimal size of an actual spacer) were obtained for each spacer. The generated $k$-mers and their reverse-complements were used as text search queries to find exact matches in $n$ viral genomes and in host genomes (self-matches). The source arrays were excluded from the $k$-mer search because these are trivially expected to produce at least one self-match. If any $k$-mer from a spacer matched the target sequence, the spacer counted as having a match for the given $k$ value.

The fraction of spacers with matches was calculated, using pseudo-counts, as $\frac{m+1}{N+2}$, where $m$ is the number of spacers with matches for a given $k$-mer size, and $N$ is the total number of spacers.

For the data acquired from adaptation experimental results, unique spacers sequences were used to generate the mock spacer set and for $k$-mer generation. For the calculation of the fraction of spacers with matches, each match of a real spacer was multiplied by its count in the original (redundant) dataset. The mock spacer set was taken as is, with each spacer occurrence counted as one.

**Statistics and reproducibility**. Statistical robustness of the comparisons was estimated by 1000-fold bootstrap resampling of the set of spacers (real, mock or scrambled) and recording the range of variation of the frequencies with the resampled sets (Supplementary Figs. 1, 3, and 4). Non-overlap of the ranges implies a $p$ value $\ll 10^{-6}$.

The results should be completely reproducible given an identical set of sequences.

**Reporting summary**. Further information on research design is available in the Nature Research Reporting Summary linked to this article.

## Data availability

The data used in the study are available from the previous publication[7] and at the NCBI FTP site (ftp://ftp.ncbi.nih.gov/pub/wolf/_suppl/spacers2020/).

## Code availability

The code used to process the data used in the study is available at the NCBI FTP site (ftp://ftp.ncbi.nih.gov/pub/wolf/_suppl/spacers2020/) and is also published at GitHub[32].

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

## Acknowledgements

The authors thank Dr. Sofiya Medvedeva for expert technical help and Koonin group members for many useful discussions. S.A.S., Y.I.W., and E.V.K. are supported by the Intramural Research Program funds of the National Institutes of Health of the USA (National Library of Medicine); research in the K.V.S. laboratory was supported by National Institutes of Health Grant GM104071 and Russian Science Foundation grant 19-14-00323. This paper is dedicated to the memory of Dr. Ekaterina Savitskaya who passed away while the manuscript was in preparation.

## Author contributions

Y.I.W. and E.V.K. initiated the project; S.A.S. performed research; S.A.S., Y.I.W., E.S., K.V.S., and E.V.K. analyzed the data; S.A.S. and E.V.K. wrote the manuscript that was edited and approved by all authors.

## Competing interests

The authors declare no competing interests.
