## [Peer Review File · Communications Biology]

Reviewers' comments:

Reviewer #1 (Remarks to the Author):

In the manuscript "Mapping CRISPR spaceromes onto prokaryotic viromes", the authors extend their analysis of CRISPR spacers from their 2017 paper "The CRISPR Spacer Space Is Dominated by Sequences from Species-Specific Mobilomes". The 2017 paper concluded that while most spacers have no known matches, they most likely target unsequenced mobile genetic elements related/similar to sequenced MGEs.

Here, they revisit the question of spacer origin and put forward 3 hypotheses as to the origin of unmapped spacers: 1) Spacers are partial matches to sequenced phages 2) Spacers are from unsequenced members of the mobilome and 3) spacers are mutated self-matches. By generating two sets of spacers, real spacers from arrays with at least one viral match and "mock" spacers sampled from the hosts own genome, they use k-mer matches to evaluate the potential that spacers are partial matches to self or virus. This approach recapitulates their previous analysis: self-targeting is strongly avoided and known spacers match viruses, with the inclusion of partial matches slightly increasing the fraction of viral matches. This leads them to conclude that most spacers originate from the unsequenced host-specific mobilome.

The question of spacer origin is interesting and important in the field of phage-microbe co-evolution, and the observation that most spacers do not match any known genetic element is also interesting and important. My main concern with this paper is that it does not do much to extend what has already been observed regarding the origin of CRISPR-Cas spacers. This paper would benefit from the addition of experiments or analyses that extend on what has been observed here and previously.

Specific comments:

The linked virome:

Which viral database was used, and does it include prophages? Why was only the linked virome, instead of the total virome used, given that different species of bacteria can share virus specific spacers? Why were plasmids and other MGEs excluded from this analysis? If the goal of the paper is to examine the origin of spacers, the inclusion of all these MGEs seems relevant. The manuscript would benefit from addition of these details to the main text and the methods.

The use of "mock" spacers derived from host genome:

I didn't fully understand the logic of using the host genome as the source of mock spacers. Is it supposed to represent the null expectation of spurious matches? This is confusing because the authors are also testing the hypothesis that the host genome is source of mutated spacer matches, so this seems circular. It would be helpful if the authors explained the use of this dataset and the rationale behind choosing it further in the text.

The authors note that integrated proviruses and MGEs in the host genome cause overestimation of the spurious match rate, and repetitive MGEs also cause issues of redundant mapping. If "mock" spacers target real integrated viruses just like real spacers do they don't seem like a good negative control. Could the authors use shuffled sequences or core genomes as the source "mock" spacers? It seems like this was an especially large issue in the Escherichia, Mycobacterium, and Bacillus datasets, making these analyses very difficult to interpret. These analyses in particular would benefit from a more thorough treatment. Specifically in the case of E. coli, it has been predicted that the main function of the E.coli CRISPR-Cas system is to regulate gene expression via self-matches (Bozic et al. 2019). It would be interesting to re-examine that hypothesis in light of these analyses, but the high rate of matches in the mock spacer dataset due to proviruses makes it difficult to evaluate the null expectation.

I also did not understand why the fraction of self-matching sequences for the mock spacers is not always 100% given that the mock spacers were generated from self sequence. Was the mock spacer "protospacer" excluded from the search? If so, this should be clarified in the main text and the methods.

Laboratory evolution with *E. coli* and coliphages T5 and lambda: I appreciated that the authors included this experimental system as it provided an interesting and useful comparison between short-term laboratory evolution experiments and bacterial-phage coevolution in the wild. One notable difference between these two frameworks is the prevalence of self-targeting spacers seen in the experimental system compared to the bulk Enterobacteriales analysis. The authors conclude that self-targeting spacers are gradually removed by purifying selection. Self-targeting in the absence of an anti-CRISPR or Cas mutation has been shown to be highly toxic and immediately lethal in many experimental systems, and would be expected to decrease microbial viability in the short term, not gradually over time. To address this discrepancy, can the authors comment more on the nature of the experimental self-targeting? Are these self-targeting events ameliorated by mismatches or incorrect PAMs? If the matches are perfect, it seems likely that the DNA is coming from dead cells, which changes the interpretation of these data.

Conclusions: The main conclusion of this work is that we are missing knowledge about the bulk of the global virome. Can CRISPR spacers/match rates to estimate how many more viral genomes would need to be sequenced to fill that gap? Conversely, can the authors assess how complete our knowledge of the global "spacerome" is?

Reviewer #3 (Remarks to the Author):

The work describes the search for the origin of CRISPR spacer sequences through the use of variable length partial matching through k-mers. To test hypotheses on the origin of the spacers, spacer collections with protospacer matches were utilized. Mock spacers were generated from genomes from which the spacers came and k-mers (8-22mers) were generated. These k-mers were used to seek for "protospacer" matches in self-genome and viral sequences as a function of k-mer length. By this method, the authors concluded that longer k-mers provide more specificity and also decrease the number of matches found in the tested databases. Through these results the authors conclude that the majority of protospacers belong to the unsequenced mobilome.

I would propose the major strength of this manuscript lies in the method of study (partial k-mer matching) rather than its biological conclusion which is the predominant view in the field. The major weakness of this manuscript is that the method is opaque and lacks most information required to replicate the findings or apply the approach to new genome sets. I also have concerns about the k-mer size range studied. While 22 was chosen as the max k-mer size, this was because of the smallest spacer size. Understanding the size distribution of the spacers would be key to interpret these results as a 22-mer representing a ~35bp spacer leaves a lot of room for matching due to chance. Some statistical analysis of the probability of finding these sequences should be included and I would propose longer k-mers should be included even if they can no longer be used to match short spacers. I would propose with additional analysis and improvements in the clarity and transparency of analysis, this manuscript would be an excellent candidate for publication in *Communications Biology*.

Specific comments:

1. A large number of CRISPR-Cas systems utilize PAM sequences to distinguish self from foreign yet the study does not consider the utility or significance of these sequences. How would the inclusion of the PAM sequence when searching for matches to the mock and real spacers change the results?

2. The study relies on the creation and analysis of K-mers of different sizes, why the exclusion of longer more specific K-mers? Presumably the majority of spacers are >>22bp and the analysis could be spread over a longer k-mer range.
3. In line 78 it is mentioned that identifiable protospacers with at least one mismatch were used for the study. The method for identifying spacers and the stringency requirements need to be clearly stated: ex was 22 the smallest allowable spacer size?
4. Lines 91-92: "oligonucleotide statistics of a spacerome is similar to that of the corresponding host genome and the linked virome". This should be elaborated on.
5. Lines 116 to 120: for the proposal of the anti-CRISPR proteins or inactivated CRISPR-Cas systems as the reason of existence of genomes with high self-targeting spacers (in prophage regions). An analysis looking for anti-CRISPR protein homologues to expand on this hypothesis would be warranted to support this claim.
6. Line 133 mentions the generation of scrambled spacers, this should also be stated in the methods with an explanation of how it was done. As it stands, it is impossible to understand the nature of the scrambling. The data generated by this method is shown in Supplementary Figure 1, the legend of the figure should reflect that the spacers were not only mock but also scrambled.
7. In line 156 the set of hosts and viruses pairs to be further analyzed are presented. The authors should include a list with the extracted genomes for this analysis.
8. In line 181 the order Enterobacteriales is mentioned, the authors should state the members of the order (as mentioned before). A supplemental table with the genome's accession data should be provided.
9. The sentence starting at line 191 ("In summary...") is misleading. The analysis of the K-mers, in this study, has allowed us to conclude that the origin of the majority of the spacers is yet unknown.
10. The statement "most of the spacers are not decaying" in line 194 is somewhat confusing. These results are informed from the K-mers used and is also biased by the input data (where only one mismatch was allowed).
11. The Materials and Methods part is lacking many important details as to how the analysis was performed and what tools were used. New tools and/or code should be made available.
12. In line 348, reference 29 is to the database but there is no information on how the viral genomes were extracted.
13. Has the code utilized for this paper been made available?
14. Line 60: MGE should be defined the first time
15. Figure 2 seems to be split into 3 separate documents rather than one with letters denoting the panels
16. Figure 1A is easily understood from the text. Including additional information (such as size distribution and/or spacers/genome distribution) might be more helpful.
17. Figure 3 should include sample size (number of genomes) as it is not intuitive if the sampling of diversity affects these results
18. The predominant method of visualizing the data (fraction vs k-mer size) is not always intuitive and its interpretation should be more clearly explained in text.
19. Many figures also support using larger k-mer sizes as saturation has frequently not occurred (see Fig 3 Sulfolobus)
20. From a computational perspective, why were canonical k-mers not used and matched to indexed genomes? Would this not drastically help scale the approach?

Reviewers' comments:

Reviewer #1 (Remarks to the Author):

In the manuscript "Mapping CRISPR spaceromes onto prokaryotic viromes", the authors extend their analysis of CRISPR spacers from their 2017 paper "The CRISPR Spacer Space Is Dominated by Sequences from Species-Specific Mobilomes". The 2017 paper concluded that while most spacers have no known matches, they most likely target unsequenced mobile genetic elements related/similar to sequenced MGEs.

Here, they revisit the question of spacer origin and put forward 3 hypotheses as to the origin of unmapped spacers: 1) Spacers are partial matches to sequenced phages 2) Spacers are from unsequenced members of the mobilome and 3) spacers are mutated self-matches. By generating two sets of spacers, real spacers from arrays with at least one viral match and "mock" spacers sampled from the hosts own genome, they use k-mer matches to evaluate the potential that spacers are partial matches to self or virus. This approach recapitulates their previous analysis: self-targeting is strongly avoided and known spacers match viruses, with the inclusion of partial matches slightly increasing the fraction of viral matches. This leads them to conclude that most spacers originate from the unsequenced host-specific mobilome.

The question of spacer origin is interesting and important in the field of phage-microbe co-evolution, and the observation that most spacers do not match any known genetic element is also interesting and important. My main concern with this paper is that it does not do much to extend what has already been observed regarding the origin of CRISPR-Cas spacers. This paper would benefit from the addition of experiments or analyses that extend on what has been observed here and previously.

General response: We appreciate the reviewer's interest in our work but have to respectfully disagree with the following conclusion: "My main concern with this paper is that it does not do much to extend what has already been observed regarding the origin of CRISPR-Cas spacers". What we add here to the previous analyses, is specific evidence of the origin of most of the spacers from the vast, unsequenced part of the mobilome. Surely, we and others have discussed that possibility previously, and considered it plausible, but this was sheer speculation. Here we show that inclusion of partial matches only slightly increases the fraction of spacers that match the mobilome and that host protospacers are strongly avoided. Together, this evidence leaves us with one possibility only, namely, that the spacers come from the vast 'dark' part of the mobilome. Further, combined with the observation that the oligonucleotide composition of the spacers is closely similar to that of the known viruses, these findings indicate that the species-specific virome consists of related viruses that, however, are divergent enough to prevent cross recognition of spacers. Perhaps, many or even most microbiologists have believed for years that the prokaryotic viromes had this type of organization. However, there was no evidence to support this concept, and here, we present such evidence for the first time. In order to emphasize these conclusions, in the revision, we changed the title of the paper and modified the concluding paragraph of the Discussion.

Specific comments:

The linked virome:

Which viral database was used, and does it include prophages?

Response: the sequences that are classified as “Viruses” at the top of the taxonomy hierarchy in GenBank were used as the source dataset. This excludes prophages by construction.

Why was only the linked virome, instead of the total virome used, given that different species of bacteria can share virus specific spacers?

Response: first, the host annotation in GenBank often covers multispecies groups, partially mitigating the problem of the data loss due to overly narrow specification. Second, given the size of the data, one can expect spurious matches to play the role, at least with shorter k-mers. Including unrelated viruses would increase the chance of a spurious match and, therefore, decrease the signal-to-noise ratio because only a small fraction of the total virome is likely to infect any particular host. We explicitly indicate this in the revised manuscript.

Why were plasmids and other MGEs excluded from this analysis? If the goal of the paper is to examine the origin of spacers, the inclusion of all these MGEs seems relevant. The manuscript would benefit from addition of these details to the main text and the methods.

Response: The complete genome assemblies were taken as whole. Some of them included plasmids, others did not. Other MGEs that are encoded in the host chromosomes were included along with the rest of the genome data. We did not have access to any additional plasmid sequence database. We clarified these details in the text.

The use of “mock” spacers derived from host genome:

I didn't fully understand the logic of using the host genome as the source of mock spacers. Is it supposed to represent the null expectation of spurious matches? This is confusing because the authors are also testing the hypothesis that the host genome is source of mutated spacer matches, so this seems circular. It would be helpful if the authors explained the use of this dataset and the rationale behind choosing it further in the text.

Response: the mock spacers do represent the null expectation of spurious matches between the sequences derived from the host and virus genomes for the given k value. There is no circularity with respect to the hypothesis of the host origin of the spacers. On the contrary, if this hypothesis holds, i.e. the spacers do originate from the host genome, the real and the mock spacers should equally poorly match the virus genomes, so that the curves would retain the approximate 1:1 ratio for all values of k . To clarify, the following sentence was added to the text: *“The mock spacer set provides the benchmark for the spurious matches between the spacerome and the virome because the overwhelming majority of the host genome sequence is unrelated to the virus genomes.”*

The authors note that integrated proviruses and MGEs in the host genome cause overestimation of the spurious match rate, and repetitive MGEs also cause issues of redundant mapping. If “mock” spacers target real integrated viruses just like real spacers do they don't seem like a good negative control. Could the authors use shuffled sequences or core genomes as the source “mock” spacers? It seems like

this was an especially large issue in the Escherichia, Mycobacterium, and Bacillus datasets, making these analyses very difficult to interpret. These analyses in particular would benefit from a more thorough treatment. Specifically in the case of E. coli, it has been predicted that the main function of the E.coli CRISPR-Cas system is to regulate gene expression via self-matches (Bozic et al. 2019). It would be interesting to re-examine that hypothesis in light of these analyses, but the high rate of matches in the mock spacer dataset due to proviruses makes it difficult to evaluate the null expectation.

Response: we appreciate this suggestion, and scrambled spacers indeed have been analyzed, to obtain the upper bound for the true matches. See the paragraph starting with “To assess the spacer-virus match rate under relaxed criteria”. There was no need for modifications, in this case, because such analysis was part of the original manuscript.

I also did not understand why the fraction of self-matching sequences for the mock spacers is not always 100% given that the mock spacers were generated from self sequence. Was the mock spacer “protospacer” excluded from the search? If so, this should be clarified in the main text and the methods.

Response: the self-match hit was expressly excluded (“Then, occurrences of these k-mers were identified in the target viral genomes and in the corresponding host genome, excluding the source arrays” in the text). We also added the explanation to the Methods.

Laboratory evolution with E. coli and coliphages T5 and lambda: I appreciated that the authors included this experimental system as it provided an interesting and useful comparison between short-term laboratory evolution experiments and bacterial-phage coevolution in the wild. One notable difference between these two frameworks is the prevalence of self-targeting spacers seen in the experimental system compared to the bulk Enterobacteriales analysis. The authors conclude that self-targeting spacers are gradually removed by purifying selection. Self-targeting in the absence of an anti-CRISPR or Cas mutation has been shown to be highly toxic and immediately lethal in many experimental systems, and would be expected to decrease microbial viability in the short term, not gradually over time. To address this discrepancy, can the authors comment more on the nature of the experimental self-targeting? Are these self-targeting events ameliorated by mismatches or incorrect PAMs? If the matches are perfect, it seems likely that the DNA is coming from dead cells, which changes the interpretation of these data.

Response: In *E. coli*, the I-E CRISPR-Cas system is not highly active, and accordingly, the toxicity of self-targeting spacers is expected to be limited. The self-targeting spacers from these experiments are not ameliorated by mismatches more than the viral ones. We checked for PAMs, and interestingly, indeed, found that the fraction of PAM mismatches was notably higher with the self-targeting spacers than it was with the virus-targeting ones (new Supplementary File 1). Nevertheless, for both types of spacers, the correct PAM was in the majority. Thus, elimination of the self-targeting spacers by selection remains the leading explanation of the difference between the virus/self ratio between the genome derived and experimentally obtained spacers. As rightly pointed out by the reviewer, this elimination did not have to be particularly gradual, it could have been rapid, on the evolutionary scale. Accordingly, in the revision, we dropped the ‘gradually’ claim while also acknowledging the possibility that some of the arrays in the experimental data could come from dead bacteria.

Conclusions: The main conclusion of this work is that we are missing knowledge about the bulk of the global virome. Can CRISPR spacers/match rates to estimate how many more viral genomes would need

to be sequenced to fill that gap? Conversely, can the authors assess how complete our knowledge of the global “spacerome” is?

Response: the size of the virome and the spacerome are tightly linked and are – we agree with the reviewer – of great interest. An accurate estimate would require involved mathematical modeling that is beyond the scope of this work, but even more importantly, the parameter values for such models will be difficult to define. We briefly discuss this matter towards the conclusion of the revised manuscript including the naïve linear approximation.

Reviewer #3 (Remarks to the Author):

The work describes the search for the origin of CRISPR spacer sequences through the use of variable length partial matching through k-mers. To test hypotheses on the origin of the spacers, spacer collections with protospacer matches were utilized. Mock spacers were generated from genomes from which the spacers came and k-mers (8-22mers) were generated. These k-mers were used to seek for “protospacer” matches in self-genome and viral sequences as a function of k-mer length. By this method, the authors concluded that longer k-mers provide more specificity and also decrease the number of matches found in the tested databases. Through these results the authors conclude that the majority of protospacers belong to the unsequenced mobilome.

I would propose the major strength of this manuscript lies in the method of study (partial k-mer matching) rather than its biological conclusion which is the predominant view in the field. The major weakness of this manuscript is that the method is opaque and lacks most information required to replicate the findings or apply the approach to new genome sets. I also have concerns about the k-mer size range studied. While 22 was chosen as the max k-mer size, this was because of the smallest spacer size. Understanding the size distribution of the spacers would be key to interpret these results as a 22-mer representing a ~35bp spacer leaves a lot of room for matching due to chance. Some statistical analysis of the probability of finding these sequences should be included and I would propose longer k-mers should be included even if they can no longer be used to match short spacers. I would propose with additional analysis and improvements in the clarity and transparency of analysis, this manuscript would be an excellent candidate for publication in Communications Biology.

Response: we have to respectfully disagree with the statement that “a 22-mer representing a ~35bp spacer leaves a lot of room for matching due to chance”. First of all, the experiments with the shuffled spacers demonstrate that the random matches practically disappear for $k \geq 15$. Second, there is almost no penalty in decomposing a 35bp spacer into 14 22-mers as the search space increases by a factor much less than 14 (all of these 22mers overlap, most of them considerably), whereas we are talking about larger differences. Using longer k-mers would force us to exclude many spacers from the analysis and/or to adjust the statistics for the variable underlying data set, with the loss of both statistical power and clarity.

Specific comments:

1. A large number of CRISPR-Cas systems utilize PAM sequences to distinguish self from foreign yet the

study does not consider the utility or significance of these sequences. How would the inclusion of the PAM sequence when searching for matches to the mock and real spacers change the results?

Response: we can utilize this information only for the relatively few CRISPR-Cas systems for which the PAM sequence is known (even if we, indeed, expect that many do use it). Additionally, most of the known PAM sequences are relatively degenerate, so they shouldn't dramatically constrain the matches. Unless there is some non-conventional, strongly biased, relationship between the PAM and the protospacer, inclusion of PAM should be equivalent to increasing k by 1 or 2. Thus, we did not deem it necessary to include PAMs in our analyses (but see PAM analysis for the experimentally acquired spacers).

2. The study relies on the creation and analysis of k -mers of different sizes, why the exclusion of longer more specific k -mers? Presumably the majority of spacers are $\gg 22$ bp and the analysis could be spread over a longer k -mer range.

Response: Actually, there are many spacers of 22 bp. Thus, using longer k -mers would force us to exclude a substantial fraction of the spacers from the analysis and/or to adjust the statistics for the variable underlying data set, with the loss of both statistical power and clarity.

3. In line 78 it is mentioned that identifiable protospacers with at least one mismatch were used for the study. The method for identifying spacers and the stringency requirements need to be clearly stated: ex was 22 the smallest allowable spacer size?

Response: The spacer-protospacer match data was taken from [7]. There was no prior limit on the spacer length; conversely, the distribution of the lengths of the identified spacers dictated the maximum k -mer length.

4. Lines 91-92: "oligonucleotide statistics of a spacerome is similar to that of the corresponding host genome and the linked virome". This should be elaborated on.

Response: Again, full details are given in the previously published work [7]. Here, we added a clarification: "*oligonucleotide statistics (GC-content, dinucleotide and tetranucleotide distributions)*".

5. Lines 116 to 120: for the proposal of the anti-CRISPR proteins or inactivated CRISPR-Cas systems as the reason of existence of genomes with high self-targeting spacers (in prophage regions). An analysis looking for anti-CRISPR protein homologues to expand on this hypothesis would be warranted to support this claim.

Response: Search for anti-CRISPR proteins was beyond the scope of this work. The association between self-targeting and prophage-encoded anti-CRISPR proteins has been amply demonstrated in previous studies including our own recent, comprehensive analysis. We clarify and cite this new work in the revision: "*Apparently, these organisms protect themselves from the self-destructive action of CRISPR-Cas system via provirus-encoded anti-CRISPR proteins, as demonstrated by previous, extensive analyses of self-targeting in conjunction with the identification of anti-CRISPR proteins*".

6. Line 133 mentions the generation of scrambled spacers, this should also be stated in the methods with an explanation of how it was done. As it stands, it is impossible to understand the nature of the

scrambling. The data generated by this method is shown in Supplementary Figure 1, the legend of the figure should reflect that the spacers were not only mock but also scrambled.

Response: The nature of the scrambling is stated in the text at the first mention: *“the sequence of each individual spacer was randomly shuffled”*.

7. In line 156 the set of hosts and viruses pairs to be further analyzed are presented. The authors should include a list with the extracted genomes for this analysis.

Response: this information was included in the Supplementary Material.

8. In line 181 the order Enterobacteriales is mentioned, the authors should state the members of the order (as mentioned before). A supplemental table with the genome’s accession data should be provided.

Response: The requested supplementary materials were added.

9. The sentence starting at line 191 (“In summary...”) is misleading. The analysis of the K-mers, in this study, has allowed us to conclude that the origin of the majority of the spacers is yet unknown.

Response: *We have to respectfully disagree. The sentence was modified for clarity but not changed to the agnostic version suggested by the reviewer.*

10. The statement “most of the spacers are not decaying” in line 194 is somewhat confusing. These results are informed from the K-mers used and is also biased by the input data (where only one mismatch was allowed).

Response: We edited the sentence for clarity but cannot see any confusion. We examined the entire applicable range of *k*-mers, from 8 to 22 (shorter *k*-mers produce mostly spurious matches whereas inclusion of longer ones would leave out many spacers as indicated above). The extra matches to *k*-mers compared to full matches are virus-specific spacers with (multiple) mismatches. There is no confusion.

11. The Materials and Methods part is lacking many important details as to how the analysis was performed and what tools were used. New tools and/or code should be made available.

Response: No new tools were constructed for this study. The software code is available, as per the Code and data availability statements that were added to the text

12. In line 348, reference 29 is to the database but there is no information on how the viral genomes were extracted.

Response: Standard, universally available Entrez utilities were used to obtain the data. The clarification was added to the Methods

13. Has the code utilized for this paper been made available?

Response: Yes, see above

14. Line 60: MGE should be defined the first time

Response: Yes, done.

15. Figure 2 seems to be split into 3 separate documents rather than one with letters denoting the panels

Response: The 3 panels were combined within the same document.

16. Figure 1A is easily understood from the text. Including additional information (such as size distribution and/or spacers/genome distribution) might be more helpful.

Response: We agree with the reviewer that this figure in the original manuscript indeed was not as informative as desirable. We included the distributions of array lengths for the genomes with and without virus-matching spacers (new panel b) and briefly discuss it in the text.

17. Figure 3 should include sample size (number of genomes) as it is not intuitive if the sampling of diversity affects these results

Response: *this information was included.*

18. The predominant method of visualizing the data (fraction vs k-mer size) is not always intuitive and its interpretation should be more clearly explained in text.

Response: Clarified as follows: "*In all comparisons, the fraction of spacers with at least one match of the given length k was used to quantify the similarity*".

19. Many figures also support using larger k-mer sizes as saturation has frequently not occurred (see Fig 3 Sulfolobus)

Response: As explained above, inclusion of longer k -mers did not appear to be helpful.

20. From a computational perspective, why were canonical k-mers not used and matched to indexed genomes? Would this not drastically help scale the approach?

Response: we do not understand what 'canonical' k -mers are.

REVIEWERS' COMMENTS:

Reviewer #1 (Remarks to the Author):

In the manuscript "Mapping CRISPR spaceromes reveals vast, host-specific viromes of prokaryotes" the authors use partial spacer matches to link viruses to hosts. The updated version of the manuscript includes some clarifying comments regarding their methodology. They also provided some extra biological context to partially explain the self-targeting seen in their bioinformatics and experimental datasets – namely anti-CRISPR proteins and PAM mismatches. These explanations and context are very important in interpreting CRISPR self-targeting. Last, they used their analyses here to speculate about the size and "shape" of the total virome. These inclusions have strengthened the manuscript, which in my opinion is sufficient for publication.